# Mummified precocial bird wings in mid-Cretaceous Burmese amber

Lida Xing[1,2,*], Ryan C. McKellar[3,4,*], Min Wang[5,*], Ming Bai[6,*], Jingmai K. O'Connor[5], Michael J. Benton[7], Jianping Zhang[2], Yan Wang[8], Kuowei Tseng[9], Martin G. Lockley[10], Gang Li[11], Weiwei Zhang[12] & Xing Xu[5,*]

Our knowledge of Cretaceous plumage is limited by the fossil record itself: compression fossils surrounding skeletons lack the finest morphological details and seldom preserve visible traces of colour, while discoveries in amber have been disassociated from their source animals. Here we report the osteology, plumage and pterylosis of two exceptionally preserved theropod wings from Burmese amber, with vestiges of soft tissues. The extremely small size and osteological development of the wings, combined with their digit proportions, strongly suggests that the remains represent precocial hatchlings of enantiornithine birds. These specimens demonstrate that the plumage types associated with modern birds were present within single individuals of Enantiornithes by the Cenomanian (99 million years ago), providing insights into plumage arrangement and microstructure alongside immature skeletal remains. This finding brings new detail to our understanding of infrequently preserved juveniles, including the first concrete examples of follicles, feather tracts and apteria in Cretaceous avialans.

[1] State Key Laboratory of Biogeology and Environmental Geology, China University of Geosciences, Beijing 100083, China. [2] School of the Earth Sciences and Resources, China University of Geosciences, Beijing 100083, China. [3] Palaeontology, Royal Saskatchewan Museum, Regina, Saskatchewan, Canada S4P 2V7. [4] Biology Department, University of Regina, Regina, Saskatchewan, Canada S4S 0A2. [5] Key Laboratory of Vertebrate Evolution and Human Origins, Institute of Vertebrate Paleontology and Paleoanthropology, Chinese Academy of Sciences, Beijing 100044, China. [6] Key Laboratory of Zoological Systematics and Evolution, Institute of Zoology, Chinese Academy of Sciences, Beijing 100101, China. [7] School of Earth Sciences, University of Bristol, Bristol BS8 1RJ, UK. [8] Institute of Geology and Paleontology, Linyi University, Linyi 276000, China. [9] Department of Exercise and Health Science, University of Taipei, Taipei 11153, China. [10] Dinosaur Tracks Museum, University of Colorado Denver, Denver, Colorado 80217, USA. [11] Institute of High Energy Physics, Chinese Academy of Science, Beijing 100049, China. [12] P.O. Box 4680, Chongqing 400015, China. * These authors contributed equally to this work. Correspondence and requests for materials should be addressed to L.X. (email: xinglida@gmail.com) or to R.C.M. (email: ryan.mckellar@gov.sk.ca).

The mid-Cretaceous Burmese amber deposit of northeastern Myanmar is one of the most prolific and well-studied sources of exceptionally preserved Mesozoic arthropod and plant fossils, but work on feathers from this deposit has just begun[1–4]. Previous studies of plumage in Cretaceous amber have been based on isolated feathers, leaving taxonomy of the feather-bearers open to debate[5,6], and amber in vertebrate bone beds has seldom yielded fossils[7]. Otherwise, Cretaceous feathers are commonly known from carbonaceous compression fossils[8–10], and three-dimensional preservation in amber is extremely rare. The combined fossil record of amber and compression fossils has provided many insights into how the feather types associated with modern birds developed[11,12], but these glimpses are restricted by preservation in each fossil type. The discovery of two partial bird wings in Burmese amber unites taxonomic and ontogenetic information from osteology with microscopic preservation down to the level of individual feather barbules and their pigment distributions. This new source of information includes integumentary features incompletely known in the compression fossil record[13].

The studied specimens come from the Angbamo site, Tanai Township, Myitkyina District, Kachin Province of Myanmar. A combination of biostratigraphy and radiometric dating have established an age estimate of 98.8 ± 0.6 Ma for this deposit[2,14,15]. The two partial wings (DIP-V-15100 and DIP-V-15101) are tiny, and are preserved within a few cubic centimetres of amber. They were examined by combining synchrotron X-ray micro-CT data for osteology, with standard macro- and microscopic observations of integumentary structures. The small size and poorly-defined articular facets indicate that both specimens were juveniles at the time of death. Although the specimens are similar in gross morphology, proportions and some plumage characteristics, their immaturity limits detailed comparisons. We tentatively suggest that the specimens belong to the same species, and suggest that the following anatomical description should be treated with some caution, given the potential for large-scale ontogenetic changes.

## Results

**Osteology.** In both wing fragments, the ulna and radius are incomplete and missing their proximal parts (Fig. 1). The ulna is mediolaterally compressed and bears a well-defined semilunate trochlear surface distally. There is no sign of quill knobs for the attachment of the secondary remiges. The radius is straight and measures less than half the width of the ulna. Only three digits are present, and the preserved phalanges suggest that the manual formula is 2-3-1. To avoid confusion about digital identities among three-fingered theropods including birds (I-II-III or II-III-IV; see ref. 16 and references therein), we use the alular, major and minor digit to refer to the anterior, middle and posterior digits, respectively. The alular metacarpal is rod-like and long, approaching one-quarter of the major digit in length, and >55% of the length of alular phalanx-1, whereas in basal avialans and non-avialan theropods, like dromaeosaurids, the latter ratio is higher[17–21]. As in most basal avialans[21,22], alular phalanx-1 fails to reach the distal end of the major metacarpal (Fig. 1b,e), but the opposite is true in most non-avialan theropods, such as dromaeosaurids and scansoriopterygids[17,23]. The alular ungual is slightly smaller than that of the major digit, and both claws are strongly recurved. The major metacarpal is robust and rod-like, and forms a weakly defined ginglymoid articular facet for major phalanx-1. As in *Sapeornis* and more advanced birds such as enantiornithines and ornithuromorphs[21,22,24], major phalanx-1 is longer than major phalanx-2; in contrast, the latter phalanx is longer in most other non-avialan theropods and basal avialans,

including *Archaeopteryx*, *Jeholornis* and *Confuciusornis*[18–20,25]. The minor metacarpal terminates distal to the major metacarpal, a synapomorphy of enantiornithines[20,26]. The minor metacarpal is dorsoventrally expanded, in caudal view appearing as robust as the major metacarpal, a feature common in enantiornithines[27]. Both specimens preserve a single, reduced, wedge-shaped phalanx in the minor digit as in some ornithothoracines.

**Plumage in DIP-V-15100.** Specimen DIP-V-15100 (Fig. 2, Supplementary Figs 1 and 2) preserves a range of plumage from the manus and distal forearm. This includes the basal portions of nine highly asymmetrical primary flight feathers and five secondary feathers (six of the primaries are conclusive, but the distinction between primary and secondary feathers is obscured by overlap within the proximal part of the wing; Figs 1a and 2c). The rachises within primaries are sub-cylindrical in cross-section (expanded dorsoventrally and constricted laterally), with a weakly developed ventral ridge and groove combination (Fig. 2d). Each barb ramus is lanceolate and bears an asymmetrical arrangement of proximal barbules that are blade-shaped throughout most of their length, along with blade-shaped distal barbules that taper into a poorly-defined pennulum bearing hooklets (Fig. 2e). The alula has three feathers clearly visible, and is distinct from the main surface of the wing (Fig. 2a,g). Barbs within the alula have much broader and more asymmetrical barbules than in the primaries and coverts, distal barbules have a well-developed

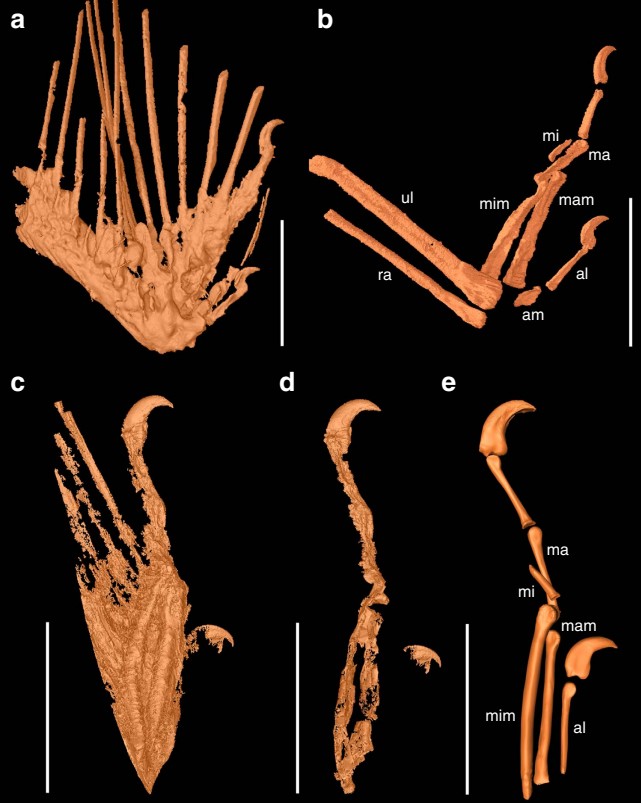

**Figure 1 | SR X-ray μCT reconstructions of osteology in DIP-V-15100 and DIP-V-15101.** (**a**) Mummified DIP-V-15100, showing rachises, skin, muscle and claws. (**b**) Skeletal morphology of DIP-V-15100, using different density threshold. (**c**) Mummified DIP-V-15101, showing rachises, skin, muscle and claws. (**d**) Skeletal morphology of DIP-V-15101. (**e**) Reconstruction of osteology based on the CT data. al, alular digit; am, alular metacarpal; ma, major digit; mam, major metacarpal; mi, minor digit; mim, minor metacarpal; ra, radius; ul, ulna. Scale bars, 5 mm.

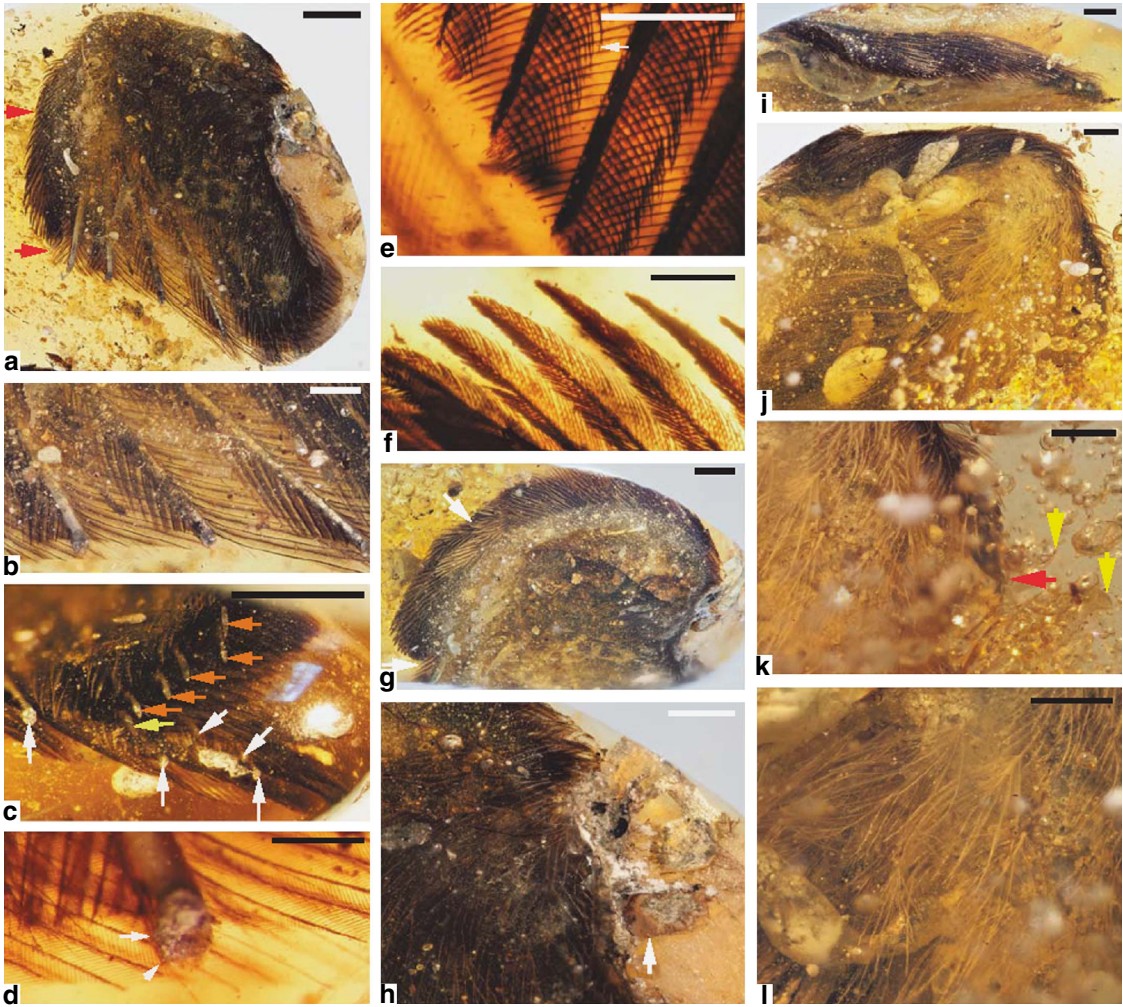

**Figure 2 | DIP-V-15100 photomicrographs. (a)** Overview of dorsal wing surface, with claws (red arrows), dorsal details in **a–h**. (**b**) Primary feathers and barbs truncated by amber surface. (**c**) Primaries (white arrows), secondaries (yellow arrows) and indeterminate feather (orange arrow), in basal zone of overlap (slightly oblique view). (**d**) Detail of primary rachis, with weak ventral ridge (inclined arrow) and barb ramus attachment somewhat low on side of rachis (horizontal arrow). (**e**) Primary feather microstructure and pigment distribution (t.l.), with hooklets on distal barbules (arrow). (**f**) Alula barbs with blunted apices and blade-like barbules with banded pigmentation. (**g**) Anterodorsal view highlighting alula separation from wing surface (arrows), as well as overall colour patterning (light hitting bubbles in amber creates a bright band paralleling edge of wing). (**h**) Bone and integument breaching amber surface, with well-preserved osteon complexes (circle of mottled bone at arrow), while most voids in bone and tissue have been permeated by milky amber, and skin is reduced to a translucent film not visible at this scale. (**i**) Extent of gas vacuoles and milky amber emanating from dorsal and ventral surfaces of wing, in anterior view. (**j**) Contrast between coverts and ventral coat of down and contours, with ventral details in **j–l**. (**k**) Current position of claw (red arrow) and claw marks within flow lines (yellow arrows). (**l**) Bases of down feathers attached to apterium with preserved skin texture and signs of saponification. Scale bars, 2.5 mm (**a,c**); 1 mm (**b,h–l**); 0.5 mm (**d**); 0.25 mm (**e,f**); 1.5 mm (**g**); t.l., transmitted light.

ventral tooth and the barbs taper to a less acute point apically (Fig. 2f). At least three rows of contour feathers are visible in the covert series, but coverage of the primary feather bases appears reduced, either as a result of weakly developed or pale major coverts, soft tissue covering the primaries, or taphonomy.

The ventral surface of the DIP-V-15100 wing (Fig. 2i,j) is partly obscured by a network of large bubbles that emanate from the surface of the wing and are heavily clouded as a result of either decay products or moisture interacting with the surrounding resin[28]. The dense mat of dark under marginal covert feathers in the prepatagium is followed posteriorly by a series that includes two rows of pale ventral coverts, then a narrow apterium that extends from the forearm into the hand, and a field of sparsely distributed down feathers (Fig. 2j–l). Details of the coverts adjacent to the primary and secondary feather bases are unclear, due to the pale colour of these feathers, combined with the

thickness of turbid amber above them. Towards the base of the wing, isolated contour feathers are preserved, but it is unclear whether these stem from the wing surface or the side of the thorax. Bi-directional claw marks within the amber flow lines, along with the abundance of decay products in the surrounding resin and the saponified appearance of exposed tissue in the apterium (Fig. 2k,l), suggest that this specimen may have been at least partially engulfed in resin while still alive, and that much of its decay took place under anaerobic conditions (see also Methods and Supplementary Discussion for taphonomy).

Preserved feather colour in DIP-V-15100 appears dark brown in the alula, and is slightly paler in the primaries and secondaries due to reduced pigmentation in the rachises, rami and basal parts of barbules. Dorsal contour feathers are generally dark in colour, but those basal to the alula may have been pale or white, and this pale patch includes some of the primary coverts (Fig. 2a,g).

Ventrally, the surface of the wing has a strong contrast between white or pale contour feathers and down adjacent to the dark brown contour feathers along the anterior margin of the wing.

**Plumage in DIP-V-15101.** In DIP-V-15101, plumage observations are limited by the thickness of the surrounding amber and the fact that many of the remiges overlap extensively (Fig. 3, Supplementary Fig. 3). This wing appears to have been disassociated from the remainder of a corpse either through decay prior to resin contact, or extensive resin flows transporting the wing away from the body (see Methods for taphonomic analyses). The lack of substantial decay products or struggle marks within the amber may also suggest an alternative, ethological explanation for the inclusion: a predator may have dismembered the wing, and discarded it to avoid consuming feathers. Aside from the truncated apices of some flight feathers, the wing and its bones are not exposed at the surface of the amber—but they are accompanied by cross-sections through sheets of feather-bearing skin that appear to represent a predominantly plumulaceous basal portion of the wing membrane (Fig. 3f,j–l). The trailing edge of the wing has been strongly deflected anterad by resin flows,

providing a view of secondary coverts on the dorsal and ventral surfaces. Although this is not an adult specimen, the relative lengths of the dorsal and ventral coverts appear to be short, like those of modern avialans—as opposed to the elongate condition that has been debated for taxa such as *Archaeopteryx* and *Anchiornis*[29,30]. Even among the secondary feathers, more than one-half of the feather is unsupported by coverts (Fig. 3a,d).

Where visible, feather structure and arrangement are consistent between the two wing specimens. DIP-V-15101 appears to contain nine primary flight feathers in the zone of feather overlap, and there are traces of at least five secondaries plus a mass of secondary coverts visible on the dorsal surface of the wing. Cross-sectional details of the primary flight feather rachises are only available at their distal extremes, where the rachises and rami both display a deep, blade-like morphology (Fig. 3g): this profile is exaggerated by the angle of the section. Unlike DIP-V-15100, the ventral coverts surrounding the bases of the primaries and secondaries are visible, and both sets of coverts are markedly shorter than the flight feathers (Fig. 3a). Barbule microstructure is difficult to discern except in feathers at the extreme margins of the wing, or associated with the sheet of skin preserved basal to the main inclusion. Covert microstructure is

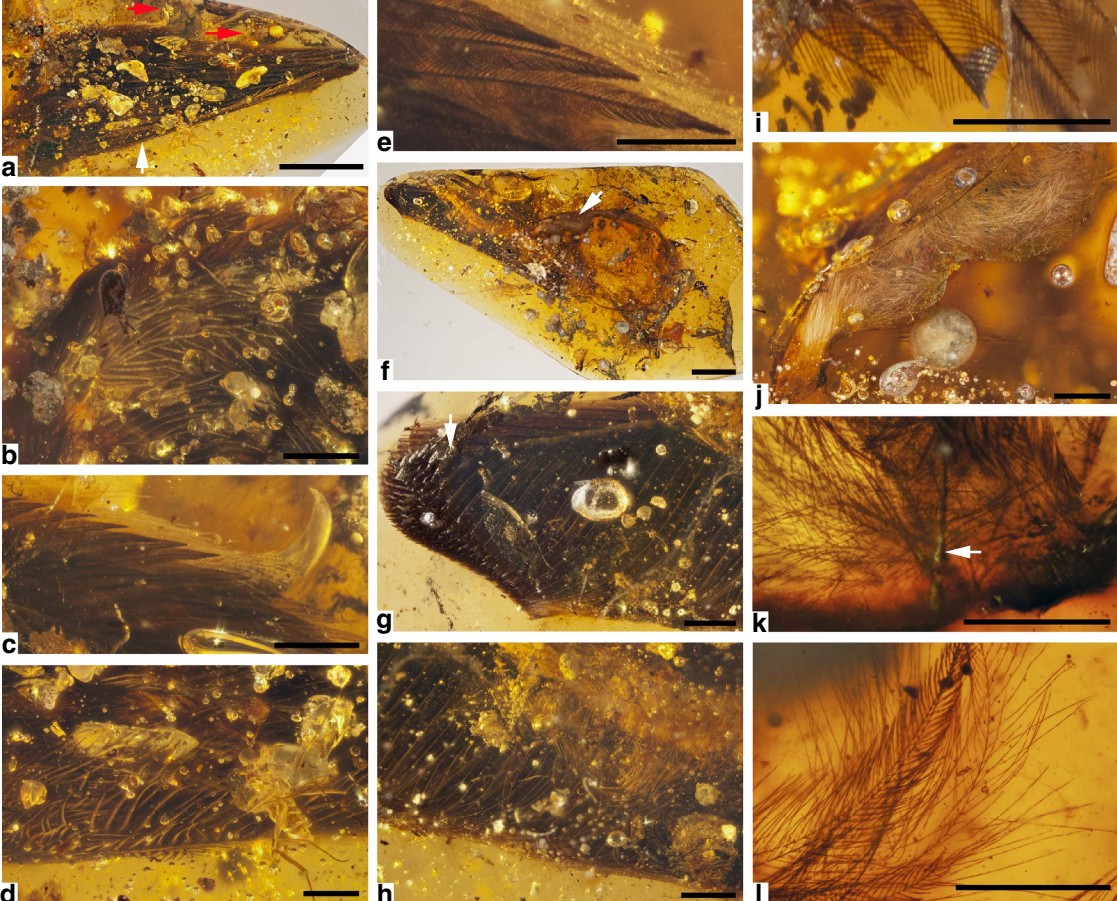

**Figure 3 | Photomicrographs of DIP-V-15101.** (**a**) Overview of dorsal wing surface, with claws (red arrows), and extent of coverts along posterior edge of wing (white arrow), dorsal details in **a–e**. (**b**) Pale or white plumage spot at base of alula. (**c**) Contrasting plumage colours and structure near alular digit and claw. (**d**) Narrow, flexible barb morphology and paler pigmentation in secondary coverts (near arrow in **a**). (**e**) Pigment distribution within blade-shaped barbules of alula. (**f**) Ventral wing surface (apex of primaries in upper left corner), with flap of feather-bearing skin trailing off the proximal edge of wing in counterclockwise direction (arrow), ventral details in **f–l**. (**g**) Primaries, where they were apically truncated by amber polishing (arrow). (**h**) Secondaries, where they have been curled by resin flows, displaying their flexible barbs, and the mixture of contour feathers and down that protrude through a veil of milky amber towards the middle of the wing surface. (**i**) Barbule morphology and pigmentation in an isolated flight feather. (**j**) Mat of white plumulaceous barbs (down) near proximal margin of skin flap, with skin towards bottom of image. (**k**) Definitive down inserting into skin surface, with calamus (arrow) and small sheath basally. (**l**) Pennaceous and plumulaceous barbs from contour feathers on the skin flap. Scale bars, 2.5 mm (**a,f**); 1 mm (**b–d,g,h,j**); 0.5 mm (**e,i,k,l**).

similar in both wings (Fig. 3d), and detached flight feathers in DIP-V-15101 display the same subtle barbule asymmetry and pennulum development, but proximal barbules are closer in shape to distal barbules, and details at the scale of barbule hooklets are not visible (Fig. 3i). Definitive down is present within the skin section preserved, with rapidly tapering rachises, obvious calami and at least one feather partially sheathed (Fig. 3j,k). Nearby, contour feathers differentiated into plumulaceous bases and pennaceous apices are visible (Fig. 3l), as are sheets of skin with feathers arranged in tracts (Supplementary Fig. 4g).

**Colour patterns**. Despite differences in the wing sizes, some plumage colouration similarities exist at both the scale of the entire wing and that of individual barbs. Specimen DIP-V-15101 shares a pale or white spot among the feathers just basal to its well-developed alula (Fig. 3a,b), and much of the paler plumage within the wing seems to be achieved through reduced pigmentation in barb rami and the bases of barbules. Unlike the smaller wing, better-defined bands of pale feathers extend across the dorsal surface of DIP-V-15101, posteriorly and distally from the apex of the alula, and along the trailing edge of the wing. The lighter colour of these feathers may be in part due to paler feather margins, particularly along the inner vane. Details of the ventral wing surface are partially hidden by decay products and inclusions in the amber, but the base of the alular digit is clearly surrounded by white under marginal coverts (Fig. 3c), and this plumage continues posteriorly, as a mixture of pale or white plumulaceous feathers that protrude through a veil of milky amber (Fig. 3f,h). Overall, the ventral wing plumage appears darker in DIP-V-15101, but this may be an artefact of feather overlap and preservation.

## Discussion

Both amber specimens contain the incomplete remains of juvenile birds, and thus preserved morphology may not accurately reflect phylogenetic position. The majority of preserved osteological features suggest that the wings belonged to a relatively derived avialan (more derived than *Archaeopteryx*, *Sapeornis*, or *Confuciusornis*). The specimens can be confidently placed within Paraves based on the following skeletal features: only three digits are present; the major digit bears highly asymmetrical feathers; and major digit-1 is longer than the subsequent phalanx. More importantly, the presence of a minor metacarpal longer than the major metacarpal is a synapomorphy for Enantiornithes. Placement within this clade may also explain the presence of fully developed feathers alongside juvenile bones. DIP-V-15101 is approximately the same size as an Early Cretaceous enantiornithine embryo (based on the length of the major digit-I, II) that was recovered with precocial feather sheets[31]. Meanwhile, DIP-V-15100 is smaller than a juvenile enantiornithine (based on the length of the minor metacarpal and phalanx) that was recovered with well-developed plumage and elongate rectrices[32]. If the amber wings do belong to enantiornithines, they add support to the concept of precocial or superprecocial juveniles within this group. Despite the small size of the individuals, none of the feathers observed in either amber specimen have extensive sheaths, or are furled in a way that would suggest immature moults[11]. The degree of feather asymmetry, barbule interlocking and the rachis profiles observed in both specimens are all consistent with feathers used in powered flight. These findings are consistent with recent work on preserved soft tissues and plumage within larger compression fossils of an enantiornithine wing from the Early Cretaceous of Spain[13]. Where visible, the angles between the rachis and the barbs (leading ~29°, trailing ~40°) in the amber specimens are comparable to those of advanced flying birds, as opposed to taxa basal to Enantiornithes[33]. However, the primary feather rachises are narrow (Fig. 2b–d), and the preserved feather lengths and skeletal material are too incomplete for a full flight capability comparison between the fossil material and modern birds[34]. At this point, the primary insight gained from the mummified amber wings is that most of the feather types found in modern avialans were likely also present in Enantiornithes, with comparable feather arrangement, pigmentation and microstructure. Fully assessing the function of this precocial plumage in hatchlings will require additional and more complete specimens.

## Methods

**Specimens studied and terminology**. The two amber pieces in this study were collected from the Angbamo site in 2015, and were polished by local miners before the lead author was able to first examine them. For each piece, the resulting finished dimensions are ~21 × 16 × 6 mm, with a mass of 1.6 g for DIP-V-15100; and ~50 × 25 × 11 mm, with a mass of 8.51 g for DIP-V-15101.

All specimens were left in their original (polished) state, but DIP-V-15101 had to be trimmed to obtain clearer views. A swarm of dipterans was removed from the posterior margin of the wing with a razor saw, and a flat surface was polished into the specimen for clearer observation. To reduce the number of bubbles obscuring the ventral surface of this wing, it was also necessary to thin the amber encasing this side of the wing, through the use of a rotary polishing wheel, various abrasive papers and wet-polishing compounds.

Wing and feather morphological terms follow those of Lucas and Stettenheim[11]. Institutional abbreviations used in the text include DIP (Dexu Institute of Palaeontology, Chaozhou, China); RSM (Royal Saskatchewan Museum, Regina, Saskatchewan, Canada). All specimens with DIP prefixes in their specimen numbers are deposited within the publicly accessible amber collection of the Dexu Institute of Palaeontology. All specimen measurements were taken with an ocular micrometre, and figures with the abbreviation t.l. refer to the use of transmitted light on a compound microscope, to record pigment distribution and finer details of feather microstructure. Osteological measurements taken from SR X-ray μCT reconstructions are presented in Supplementary Table 1.

**Specimen microtomography**. Specimen DIP-V-15100 was imaged using propagation phase-contrast synchrotron radiation microtomography (PPC-SR-μCT) on the beamline 13 W at the Shanghai Synchrotron Radiation Facility (SSRF). The isotropic voxel size is 3.25 μm. The beam was monochromatized at an energy of 25 keV using the double crystal monochromator. To obtain a phase-contrast effect, we used a sample-detector distance (propagation distance) of 300 mm and 1,600 projections on 180°. The phase retrieval and slice reconstruction were performed using PITRE software[35]. The amber specimen DIP-V-15101 was scanned with a MicroXCT 400 (Carl Zeiss X-ray Microscopy Inc., Pleasanton, USA) at the Institute of Zoology, Chinese Academy of Sciences. The scan was done with a beam energy of 60 kV, 133 μA, absorption contrast and a spatial resolution of 4.8022 μm.

Based on the obtained image stacks, three-dimensional structures of the specimen were reconstructed and virtually dissected with Amira 5.4 (Visage Imaging, San Diego, USA). The subsequent volume rendering and animation were performed with VG Studiomax 2.1 (Volume Graphics, Heidelberg, Germany). For selected illustrations, parts of the animal (for example, the skin, feathers and so on.) were virtually removed. Final figures were prepared with Photoshop CS5 (Adobe, San Jose, USA) and Illustrator CS5 (Adobe).

**Taphonomic analysis of resin flows**. To better understand the taphonomy of the wing inclusions, to test the provenance of the amber, and to test for signs of specimen manipulation, basic observations with ultraviolet light were conducted on both specimens (Supplementary Fig. 1). Fluorescence colours observed were identical to those found within other Burmese amber specimens in the RSM collection, and match the distinctive blue fluorescence colour known for the deposit[2]. In both specimens, the wings and their decay products interacted with drying lines in the amber to an extent that precludes forgery. The uniform fluorescence colour also confirms that all parts of each specimen are original amber, and have not been cut or modified. Ultraviolet light used to observe flow lines and fluorescence within the amber was provided by a 395 nm wavelength light source, and the resulting data were captured with the same photography equipment as was used for macrophotography in this study. Visible fluorescence was stronger than it appears within the resulting photographs, due to the exposure times used in the macrophotography to capture details of amber flow lines.

Coupling ultraviolet images with those obtained through standard light photography (Figs 2 and 3) provides some insight into how each wing interacted with the surrounding resin prior to polymerization, and the nature of the decay products produced by each wing. In DIP-V-15100, the wing breaches the surface of the amber, but it appears to be contained within a single resin flow that constitutes roughly one-quarter of the amber piece's volume. The dorsal surface of the wing has a slight veil of bubbles and milky amber, but the ventral surface has numerous

large bubbles with milky surfaces (Fig. 2i,j). These ventral bubbles coalesce and are concentrated along the next major drying line in the amber, with many bubbles crossing a minor drying line in the amber along the way (Supplementary Fig. 1b,c), in a pattern that is highly suggestive of decay products or trapped air flowing out from the wing. These patterns are mirrored by bands of milky amber that are often attributed to decay products or moisture interacting with the surrounding resin[36]. Unfortunately, the claw marks that are visible within the amber (Fig. 2k) are covered by too great a thickness of amber to observe anything beyond the most prominent mark under ultraviolet lighting (Supplementary Fig. 1b). Based on the direction of decay gas flow, it seems as though the ventral surface of the wing was facing up after burial, and that the wing may have worked its way deeper into a flow of resin (on the dorsal surface) as a result of a struggle or resin deformation.

Ultraviolet images for DIP-V-15101 are less informative, because most of the wing is deeply buried within the amber, and situated in relatively thick resin flows. Where the wing does breach the amber surface (Supplementary Fig. 1g), there are clear interactions between the individual primary feathers and the surrounding resin. Resin flowed from the posterior edge of the wing towards the anterior edge—this has imparted a series of minor drying lines (Supplementary Fig. 1g) within an otherwise massive flow of resin visible on the opposite side of the amber piece (Supplementary Fig. 1h). This resin flow pattern is supported by the extent of overlap in primary and secondary flight feathers within the specimen, the strong ventral deflection seen within the trailing edge of the wing (Fig. 3g,h) and the curling pattern seen within the associated skin flap (Fig. 3f,j, Supplementary Fig. 4). Where the skin flap breaches the surface of the amber (Supplementary Fig. 1e), skin that is almost invisible using standard light microscopy (Fig. 3, Supplementary Fig. 4) forms a prominent line in the ultraviolet photograph. These two lines of observation show that the inner surface of the skin is facing the observer, forming a concave surface with many feathers branching away from the observer, but often distorted by flow lines within the amber.

**Taphonomic analysis of surface exposures.** Direct exposures of wing material in DIP-V-15100 suggest that the specimen was once part of a larger amber inclusion that may have included a greater part of the wing, or even an entire corpse. Bones, skin and feathers are all cleanly truncated at the polished amber surface, and there are no signs to suggest that the wing has been disassociated. Surface exposures within DIP-V-15100 also provide some insight into preservation quality. The bone that protrudes from the radius and ulna retains distinct osteon structures (Fig. 2h), which point towards a lack of bone recrystallization. However, the internal voids within each bone (as well as spaces between the bones and feather bases that correspond to soft tissues and the pith cavity of rachises) appear to have been infilled with grey material distinct from the larger bubbles containing decay products (Fig. 2b–d,h–j). This grey substance may be opaque amber produced as a result of interaction between the resin and fluids or decay products generated from soft tissues. However, the grey material is opaque and displays a somewhat granular texture where it is exposed, so we could not completely discount the possibility that it represents a mineral deposited within voids in the amber. Where material with a slightly more granular appearance was sampled near the radius and ulna exposures, it proved to be clay-like in composition (see scanning electron microscopy (SEM) analyses below). Where individual feathers breach the surface of the amber piece, or have a very thin layer of amber encasing them, apparent feather colour is similar to that observed in regions of extensive feather overlap.

Surface exposure for DIP-V-15101 is less substantial than in DIP-V-15100, and does not include bone exposures. This strongly suggests that the wing was disassociated from the remainder of the body, either through partial encapsulation (the remainder of the body was exposed and not preserved), dismemberment by a predator, or body components drifting apart in separate resin flows. Where the primary flight feathers are truncated by the polished surface, the pith cavity lacks the opaque amber seen in the smaller wing specimen. Milky amber and decay-related bubbles are also less prevalent around the specimen. These decay-related features may have been swept away from the wing by large-scale resin flows, or the wing inclusion may have dried and undergone decay prior to entering the resin. Given the distortions and dislocations seen within the plumage and the piece of associated skin, it seems as though resin flow has had a strong influence within this piece of amber. Where feathers are near the surface of the amber or exposed, they are noticeably darker than those in specimen DIP-V-15100. The sheet of skin preserved within DIP-V-15101 shows variable preservation. Most of the skin is translucent (nearly transparent), and small sections near the surface appear to be partially carbonized (Supplementary Fig. 4). Although many of the contour feathers are separated from the skin sheet and wing (Supplementary Figs 3 and 4), there are at least a few sections within the shallowly buried skin sheet where the arrangement of original feather rows is clearly visible (Supplementary Fig. 4f–h).

**SEM analyses of surface exposures.** SEM observations were made at the University of Alberta, Department of Earth and Atmospheric Science (Edmonton, Alberta, Canada), using a Zeiss Sigma 300 VP-FESEM operated in variable pressure mode with uncoated samples. Semi-quantitative elemental analyses were carried out using EDS.

Near the surface exposure of the radius and ulna in DIP-V-15100, a small fracture in the amber provided the opportunity to sample a flake that could potentially contain

the contact surface of the plumage and skin with the amber, some of the coarser branching structures (rachises or rami) from the coverts and the mineral or milky amber that had penetrated the gap between the wing surface and bones (Supplementary Fig. 2a). Unfortunately, once the flake was mounted for SEM observation, it became obvious that it was dominated by the latter material, with a narrow and predominantly recessive band present in the areas that might access the plumage or integument (Supplementary Fig. 5). In the region containing a thick carbonized trace of a feather rachis or ramus, the carbonized structure is barely distinguishable from the surrounding amber on the basis of its morphology. Furthermore, its chemical composition is dominated by C (Supplementary Figs 5f and 6a)—but it is difficult to ascertain whether this is exclusive to the carbon film, or if the measurement may contain contributions from the surrounding amber. Observation of the mottled grey infill was more informative from a taphonomic perspective.

Just as in the other exposed feathers in DIP-V-15100 (Fig. 2b–d), the exposed rachis or ramus central cavity in the SEM sample (as well as the region between the plumage and exposed bones) was infilled with grey material that could either represent a mineral, or strongly clouded amber. The mottled appearance of this infill suggested the former interpretation, and this was supported by both the structure and chemistry visible through SEM and EDS analysis. Unlike the carbonized feather fragment or the surrounding amber, the infill displayed a range of sheet-like structures rich in Al and Si (Supplementary Figs 5f and 6b), indicative of clay minerals. It appears as though the voids within at least some of the feather pith areas, as well as the void left behind through soft tissue decay or carbonization, was subsequently infilled by clay. There were no obvious traces of keratin sheets or melanosomes along the rachis section or in the area between the clay infill and the surrounding amber. This region was highly recessive (Supplementary Fig. 5d,e), and the only noteworthy structures in this region are the small but distinctive pyrite framboids clinging to the amber surface (Supplementary Fig. 5d). These minerals may have been the product of decay under anaerobic conditions[37].

**Photography and microscopy for structure and pigments.** Specimen macro-photography was conducted using a Visionary Digital photography station at the RSM, consisting of a Canon EOS 5D DSLR camera equipped with a Canon MP-E 65 mm Macro Photo Lens and tube extensions. Extended depth of field at high magnifications was achieved by combining multiple images from a range of focal planes using Helicon Focus 5.3.14 (Helicon Soft, Kharkov, Ukraine) software. The amber samples were photographed in their raw (polished) state, and also suspended in a glycerin bath to improve their optical characteristics.

Photomicrographs were prepared with a Sony NEX-5 camera attached to the trinocular port of an Olympus CH30 compound microscope. The amber samples were relatively thick and had to be suspended in a glycerin bath to transmit light properly. The focal distance required for observations under this setup limited microscopy to the × 4 and × 10 objective lenses. A × 40 oil immersion lens was applied to the amber samples as well, but we were unable to find any points where plumage was close enough to the flattened surfaces of the sample for clear observations.

Although both wings appear nearly black in hand sample, macro- and microscopic observation of the feathers under a range of lighting conditions indicate that DIP-V-15100 was preserved with a predominantly dark walnut brown colour, and most of DIP-V-15101 is a darker, black-brown colour. In both specimens, contour feather colours range from slightly paler variants of the main wing colour to silvery or white bands, and down feathers appear to be predominantly white in colour. Similarities in overall colour, combined with the prominent pale spot found basal to the alula suggest that the specimens may be conspecific. Without substantial broken surfaces through the feathers, it was not possible to use SEM to identify melanosomes, and so the original feather colours are speculative, even if hints of the patterns may be discerned.

Due to sample thickness, it was not possible to make detailed observations of the barbule node structures that have proven so informative in modern bird plumage identification[38,39]. Partial views of some flight feathers indicated the presence of hooklets or a differentiated base and pennulum in each amber sample or pigmentation banding that corresponds to the number of basal cells within each barbule (Figs 2 and 3), but finer morphological details were not consistently available. Where down was exposed near the surface of the amber samples, sample thickness precluded the use of dark field microscopy to examine nodes in translucent barbules, as was done in previous studies[6].

**Higher resolution images and interpretive diagrams.** To provide clearer views of the fine details within each specimen, some of the figures presented in the main text are expanded upon (Supplementary Figs 7–14) with larger-scale, high-resolution versions of the figures, and line diagrams to highlight key features. Supplementary Discussion also encompasses feather types not observed within the amber specimens, and comparisons to juvenile modern bird wings embedded in synthetic resin (Supplementary Fig. 15).

**Data availability.** The data that support the findings of this study are available from the corresponding author upon request.

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

## Acknowledgements

We thank A. Wolfe, N. Gerein and R. Poulin for discussion. This research was funded by the National Basic Research Program of China (973 Project: 2012CB822000); the Natural Sciences and Engineering Research Council of Canada (2015-00681), the National Geographic Society, USA (EC0768-15); the National Natural Science Foundation of China (No. 41402017), Scientific Research Equipment Development Project of Chinese Academy of Science (YZ201211, YZ201509), the Innovation Project of BASIC, CAS (2014-01), the National Science Fund for Fostering Talents in Basic Research (Special Subjects in Animal Taxonomy, NSFC-J1210002), the Special Fiscal Funds of Shaanxi Province (No. 2013-19), the Natural Environment Research Council (UK, NE/I027630/1, to M.J.B.) and by a Humboldt Fellowship (to M.B.). We thank Shanghai Synchrotron Radiation Facility (BL13W1) for beam time access based on proposal 15ssrf00730, and Beijing Synchrotron Radiation Facility (4W1A and 4W1B) for beam time access.

## Author contributions

L.X., R.C.M., X.X., M.B. and G.L. designed the project. L.X., R.C.M., M.W., J.K.O., M.B., M.J.B., J.Z., Y.W., K.T., G.L., W.Z. and X.X performed the research, and L.X., R.C.M., M.W., J.K.O., M.B., G.L. and X.X. wrote the manuscript.
