## [Peer review file · Nature Communications]

Reviewers' comments:

Reviewer #1 (Remarks to the Author):

This is a very nice paper on some really unusual specimens, the like of which I cannot remember seeing in the literature. The specimens are deposited in a public repository, the methods are clear, the description is well written, and the authors are cautious with their conclusions. I recommend this for publication.

I have one question: will high-resolution images of these figures be reposted in any place where they can be examined more closely? I do not say that they must be publicly available to all for republication, just that they can be examined in greater detail. thanks.

Reviewer #2 (Remarks to the Author):

A. Summary of the key results

B. Originality and interest

This interesting manuscript describes two partial wings with plumage, exquisitely preserved in mid-Cretaceous amber from Myanmar. This is the first time that fossil wings with integumentary structures and plumage are described from amber. In this sense, the claims are novel. This study shows that most of the feather types found in modern birds were likely present in Enanthionites. However, a compression fossil of an enantiornithine wing recently described from the Cretaceous of Spain (see below) also provides, in part, similar anatomical details and data. This study will be of great interest for all scientists working on early birds and evolution of feathers and plumage, and this paper undoubtedly represents an advance in the field.

C. Data & methodology: validity of approach, quality of data, quality of presentation

D. Appropriate use of statistics and treatment of uncertainties

E. Conclusions: robustness, validity, reliability

F. Suggested improvements: experiments, data for possible revision

The claims of the paper are convincing and the conclusions are robust, all being well presented. Main experiments and analysis (photomicrographs, UV photomicrographs, SEM, EDS, SR x-ray μ CT reconstructions) available nowadays have been performed for a detailed and complete description of such unique fossils. However, a table providing dimensions (length, width at mid-shaft, in mm) of various skeletal elements (complete, or as preserved for ulna and radius in XXY-V-100). It seems that this can be easily done on the basis of the osteological reconstructions presented in Fig. 1B and E. Such data, using bone robustness (width/length) and proportions between metacarpals and phalanges of the three digits, would be welcome and useful. If the two specimens belong to the same species of enantiornithines as suggested by the authors, there should be no significant differences between these

proportions.

The methods are clearly described at the end of the manuscript, and the supplementary information is complete and well presented, with useful data and figures. According to their catalogue number including the institutional abbreviation "XXY", I guess that the two specimens are curated in the Xinxiyu Institute of Palaeontology. Please include a sentence that clearly indicates (and confirms) that the Xinxiyu Institute of Palaeontology is the institutional repository of both studied specimens.

Generally, I consider that the claims are appropriately discussed in the context of previous literature. However, an important paper recently published [Navalón G, Marugán-Lobón J, Chiappe LM, Sanz JL, Buscalioni AD, 2015. Soft-tissue and dermal arrangement in the wing of an Early Cretaceous bird: Implications for the evolution of avian flight. *Scientific Reports* 5:14864] is omitted in the discussion. Navalón et al.'s paper also describes an exquisitely preserved 125-million-year-old enantiornithine wing with integumentary structures and plumage from the Las Hoyas Lagerstätte. Although it is a compression fossil, this specimen reveals the overall morphology of the integument of the wing and other connective structures associated with the insertion of flight feathers. It preserves the alular patagium, the calami of flight feather embedded in the postpatagium, and the feather follicle pattern in the propatagium. The results presented in Navalón et al.'s paper are crucial here and must be taken into account in the anatomical comparison and general discussion.

G. References: appropriate credit to previous work?

Except the missing reference to Navalón et al.'s paper and to a few papers on Cretaceous feather in the introduction (see annotated PDF), the credit to previous is appropriate.

H. Clarity and context: lucidity of abstract/summary, appropriateness of abstract, introduction and conclusions

The abstract is clear and all its content is appropriate. The introduction and conclusions are well constructed and provide the information needed here.

Reviewer #3 (Remarks to the Author):

Comments to the authors

Review: Mummified precocial bird wings in mid-Cretaceous Burmese amber

By Lida Xing et al.

The fossil record of avian feathers has relied on carbonized impressions that lack structural details. Xing et al. describe two little isolated wings in amber from the mid-Cretaceous of Myanmar, which are associated with feathers and wing bones in natural articulation. The specimens provide a wealth of anatomical and taphonomic information about the structure and arrangement of feathers in hatchlings. The authors use sophisticated synchrotron x-ray micro-CT method along with standard macro- and

microscopic observations to interpret these fossils. The paper is novel and will generate a great deal of interest to others in the field. Because this type of fossil preservation of early bird in amber is extremely rare, and the science and methodology of interpreting the fossils are sound, the manuscript is worthy for consideration in publication in Nature Communications.

However, the manuscript is not well organized; it needs careful editing and rewriting. A ms is easier to understand when the findings and significance can be quickly found because of logical and specified order presentation. In this ms, the reader has to figure out why the bird is precocial and why it is an enantiornithine. For example, if the authors state in the beginning that the wings are preserved in a space of 1 cm² amber, the readers would know instantly that it's a hatchling of a small bird (about the size of a baby finch or robin), where the wingspan would be about 3.5 cm (?). Because it's a baby bird and the bones are ossified, then the mode of development is precocial. Similarly, enantiornithine affinity is revealed from the proportions of the finger bones (long major phalanx-1, etc.). If this information is stated in the beginning, it will generate a great deal of interest.

During description of the feathers, the authors should say that four (?) kinds of feathers are preserved in the specimens: 9 primaries, (5+) secondaries, alular, and covert. Then they describe the feathers in sequence. The authors should add an extra figure showing a reconstruction of a half wing with primary and secondary feathers with rachis, barbs, and barbules.

The authors did not discuss the taphonomy of the specimens. To me this would be the most interesting part of the story. Since the wings were preserved in amber, the bird must be arboreal (enantiornithines were arboreal). The authors note: 'this wing appears to have been disassociated from the remainder of the corpse.' Why?? Perhaps the wings of the hatchlings were dismembered at the elbow joint before predation to avoid feathers, common tactics of bird predation, while the predator ate the meaty body part. The dismembered wings were then trapped in the amber.

Figures: Fig. 1 is fine. Figs 2 and 3 need rearrangement. The authors should provide simple sketches of Fig. 2a, 2b, 2c, and 2f alongside the photographs to show the detailed structures and remove some figures, which are not essential. Similar photo/sketch combination could be done for Fig. 3e, 3j, 3k, and 3l.

Author Response to Reviewers' comments:

Overall, the reviewers' comments were quite constructive and we have been able to meet all of their requests. A detailed response to each reviewer comment is provided below each point that was raised.

Reviewer #1 (Remarks to the Author):

This is a very nice paper on some really unusual specimens, the like of which I cannot remember seeing in the literature. The specimens are deposited in a public repository, the methods are clear, the description is well written, and the authors are cautious with their conclusions. I recommend this for publication.

I have one question: will high-resolution images of these figures be reposted in any place where they can be examined more closely? I do not say that they must be publicly available to all for republication, just that they can be examined in greater detail. thanks.

We have included enlarged versions of most of the figures within the expanded Supplementary Information section (Supplementary Figs. 7–14), as requested by Reviewer #3. These should cover the key elements requested by Reviewer #1. We are more than willing to add larger, high-resolution copies of all other figures to the Supplementary Information, if these are deemed necessary. As usual, we will also share images with any interested researchers, in our role as corresponding authors.

Reviewer #2 (Remarks to the Author):

- A. Summary of the key results
- B. Originality and interest

This interesting manuscript describes two partial wings with plumage, exquisitely preserved in mid-Cretaceous amber from Myanmar. This is the first time that fossil wings with integumentary structures and plumage are described from amber. In this sense, the claims are novel. This study shows that most of the feather types found in modern birds were likely present in Enanthionites. However, a compression fossil of an enantiornithine wing recently described from the Cretaceous of Spain (see below) also provides, in part, similar anatomical details and data. This study will be of great interest for all scientists working on early birds and evolution of feathers and plumage, and this paper undoubtedly represents an advance in the field.

The reviewer is absolutely correct here—we have added discussion of the recent paper that we missed to the revised manuscript (see below).

C. Data & methodology: validity of approach, quality of data, quality of presentation

D. Appropriate use of statistics and treatment of uncertainties

E. Conclusions: robustness, validity, reliability

F. Suggested improvements: experiments, data for possible revision

The claims of the paper are convincing and the conclusions are robust, all being well presented. Main experiments and analysis (photomicrographs, UV photomicrographs, SEM, EDS, SR x-ray μ CT reconstructions) available nowadays have been performed for a detailed and complete description of such unique fossils. However, a table providing dimensions (length, width at mid-shaft, in mm) of various skeletal elements (complete, or as preserved for ulna and radius in XXY-V-100). It seems that this can be easily done on the basis of the osteological reconstructions presented in Fig. 1B and E. Such data, using bone robustness (width/length) and proportions between metacarpals and phalanges of the three digits, would be welcome and useful. If the two specimens belong to the same species of enantiornithines as suggested by the authors, there should be no significant differences between these proportions.

We have added the requested table of dimensions (the new Supplementary Table 1). There are differences between the two specimens, but we do not believe that these would justify treating the specimens as two distinct species. These measurements are susceptible to differences in the reconstruction (the density thresholds utilized can affect the thickness of the reconstructed elements), and given that these appear to be hatchlings of different size, the measurements may be affected by allometric growth.

The methods are clearly described at the end of the manuscript, and the supplementary information is complete and well presented, with useful data and figures. According to their catalogue number including the institutional abbreviation "XXY", I guess that the two specimens are curated in the Xinxiyu Institute of Palaeontology. Please include a sentence that clearly indicates (and confirms) that the Xinxiyu Institute of Palaeontology is the institutional repository of both studied specimens.

We have added a sentence regarding the institutional repository to the methods section. In this case, we chose to use the Dexu Institute of Palaeontology, Chaozhou, China, because this is a publicly accessible, non-profit repository that allows better access to the specimens for other researchers.

Generally, I consider that the claims are appropriately discussed in the context of previous literature. However, an important paper recently published [Navalón G, Marugán-Lobón J, Chiappe LM, Sanz JL, Buscalioni AD, 2015. Soft-tissue and dermal arrangement in the wing of an Early Cretaceous bird: Implications for the evolution of avian flight. *Scientific Reports* 5:14864] is omitted in the discussion.

Navalón et al.'s paper also describes an exquisitely preserved 125-million-year-old enantiornithine wing with integumentary structures and plumage from the Las Hoyas Lagerstätte. Although it is a compression fossil, this specimen reveals the overall morphology of the integument of the wing and other connective structures associated with the insertion of flight feathers. It preserves the alular patagium, the calami of flight feather embedded in the postpatagium, and the feather follicle pattern in the propatagium. The results presented in Navalón et al.'s paper are crucial here and must be taken into account in the anatomical comparison and general discussion.

We apologize for missing this important work, and have included it in the revised paper. Our findings are entirely consistent with those of Navalón et al. (2015), so we have added mention of this paper to both the introduction and conclusions of the revised manuscript. This work focussed largely on membranes, musculature, and flight ability within enantiornithines, based on a wing preserved as a compression fossil with traces of soft tissues. Our discussion of this work centres around flight ability and preserved structures, and how we observe many of the same features in the amber specimen. Supplementary Figure 7 provides the overview diagram requested by Reviewer 3, and it also serves as a direct comparison to the work of Navalón et al. (2015).

G. References: appropriate credit to previous work?

Except the missing reference to Navalon et al.'s paper and to a few papers on Cretaceous feather in the introduction (see annotated PDF), the credit to previous is appropriate.

This reference has been added to the revised manuscript.

H. Clarity and context: lucidity of abstract/summary, appropriateness of abstract, introduction and conclusions

The abstract is clear and all its content is appropriate. The introduction and conclusions are well constructed and provide the information needed here.

Reviewer #3 (Remarks to the Author):

Comments to the authors

Review: Mummified precocial bird wings in mid-Cretaceous Burmese amber

By Lida Xing et al.

The fossil record of avian feathers has relied on carbonized impressions that lack structural details. Xing et al. describe two little isolated wings in amber from the mid-Cretaceous of Myanmar, which are associated with feathers and wing bones in natural articulation. The specimens provide a wealth of anatomical and taphonomic information about the structure and arrangement of feathers in hatchlings. The authors use sophisticated synchrotron x-ray micro-CT method along with standard macro- and

microscopic observations to interpret these fossils. The paper is novel and will generate a great deal of interest to others in the field. Because this type of fossil preservation of early bird in amber is extremely rare, and the science and methodology of interpreting the fossils are sound, the manuscript is worthy for consideration in publication in Nature Communications.

However, the manuscript is not well organized; it needs careful editing and rewriting. A ms is easier to understand when the findings and significance can be quickly found because of logical and specified order presentation. In this ms, the reader has to figure out why the bird is precocial and why it is an enantiornithine. For example, if the authors state in the beginning that the wings are preserved in a space of 1 cm² amber, the readers would know instantly that it's a hatchling of a small bird (about the size of a baby finch or robin), where the wingspan would be about 3.5 cm (?). Because it's a baby bird and the bones are ossified, then the mode of development is precocial. Similarly, enantiornithine affinity is revealed from the proportions of the finger bones (long major phalanx-1, etc.). If this information is stated in the beginning, it will generate a great deal of interest.

The reviewer is correct, and we have changed the manuscript so that these details are brought forward, into the abstract, or to be introduced much earlier within their respective sections. These changes are visible as tracked changes near lines 33, 58, and 66.

During description of the feathers, the authors should say that four (?) kinds of feathers are preserved in the specimens: 9 primaries, (5+) secondaries, alular, and covert. Then they describe the feathers in sequence. The authors should add an extra figure showing a reconstruction of a half wing with primary and secondary feathers with rachis, barbs, and barbules.

The new Supplementary Figures 7 and 8 provide all of the requested details, based directly on the specimens. This provides a conservative interpretation of the structures involved, and also allows us to provide larger, higher-resolution copies of the images involved.

The authors did not discuss the taphonomy of the specimens. To me this would be the most interesting part of the story. Since the wings were preserved in amber, the bird must be arboreal (enantiornithines were arboreal). The authors note: 'this wing appears to have been disassociated from the remainder of the corpse.' Why?? Perhaps the wings of the hatchlings were dismembered at the elbow joint before predation to avoid feathers, common tactics of bird predation, while the predator ate the meaty body part. The dismembered wings were then trapped in the amber.

Some of these details were discussed in the original manuscript, as part of the Taphonomy section of the Supplementary Information (under the headings of "Surface exposures" and "Mapping flows under UV light"). Specimen DIP-V-15100 most likely was encapsulated in amber while the animal was still alive, as suggested by struggle marks surrounding the claws in amber (discussed in the main text and SI Taphonomy sections). However, the reviewer makes an excellent point regarding DIP-V-15101. Predation was a possibility that was overlooked in the initial submission. We have taken the reviewer's advice, and added mention of this possibility to both the main text (near line 139) and SI Taphonomy

section.

Figures: Fig. 1 is fine. Figs 2 and 3 need rearrangement. The authors should provide simple sketches of Fig. 2a, 2b, 2c, and 2f alongside the photographs to show the detailed structures and remove some figures, which are not essential. Similar photo/sketch combination could be done for Fig. 3e, 3j, 3k, and 3l.

The reviewer's suggestions have been met as best as possible. Providing sketches within the main text figures is not entirely practical due to publication space constraints, so we have provided this information through the addition of Supplementary Figures 7–14. These figures are large enough to provide meaningful interpretive diagrams, while simultaneously meeting the request by Reviewer 1 to provide access to larger-format, higher-resolution images. All of the requested figures and sketches have been provided, with the exception of Fig. 3e, which was not significantly different from Fig. 2f, and was sufficiently illustrated through a large-format microphotograph. We believe that the main text figure layouts used provide a rapid overview of the various features of the specimens. These components should provide the range of coverage that is sought by the various audiences likely to read the paper, without having to rely heavily on the Supplementary Information to convey the basic results of the work.

Reviewers' comments:

Reviewer #2 (Remarks to the Author):

The new version of the manuscript now appears to be suitable for publication in Nature Communications in its present form. All my requests and suggestions have been followed. With the additions and corrections made here, the authors have greatly improved the quality of this paper.

Reviewer #3 (Remarks to the Author):

The authors addressed my critical comments when preparing the revised version of the manuscript. I recommend the paper for publication in its current form.